# Decellularized Antler Cancellous Bone Matrix Material Can Serve as Potential Bone Tissue Scaffold

**DOI:** 10.3390/biom14080907

**Published:** 2024-07-25

**Authors:** Yusu Wang, Ying Zong, Weijia Chen, Naichao Diao, Quanmin Zhao, Chunyi Li, Boyin Jia, Miao Zhang, Jianming Li, Yan Zhao, Rui Du, Zhongmei He

**Affiliations:** 1College of Chinese Medicinal Materials, Jilin Agricultural University, Changchun 130118, China; wangyusu@mails.jlau.edu.cn (Y.W.); zongying@jlau.edu.cn (Y.Z.); weijiac@jlau.edu.cn (W.C.); diaonaichao@jlau.edu.cn (N.D.); zhaoquanmin@jlau.edu.cn (Q.Z.); lichunyi@cstu.edu.cn (C.L.); zhangmiao96@mails.jlau.edu.cn (M.Z.); lijianming773@jlau.edu.cn (J.L.); zhaoyan@jlau.edu.cn (Y.Z.); 2Institute of Antler Science and Product Technology, Changchun Sci-Tech University, Changchun 130112, China; 3College of Animal Medicine, College of Animal Science and Technology, Jilin Agricultural University, Changchun 130118, China; jiaboyin@jlau.edu.cn

**Keywords:** antler, decellularized cancellous bone matrix, bone repair, biomaterials, RNA-sequencing analysis

## Abstract

Due to the limited supply of autologous bone grafts, there is a need to develop more bone matrix materials to repair bone defects. Xenograft bone is expected to be used for clinical treatment due to its exact structural similarity to natural bone and its high biocompatibility. In this study, decellularized antler cancellous bone matrix (DACB) was first prepared, and then the extent of decellularization of DACB was verified by histological staining, which demonstrated that it retained the extracellular matrix (ECM). The bioactivity of DACB was assessed using C3H10T1/2 cells, revealing that DACB enhanced cell proliferation and facilitated cell adhesion and osteogenic differentiation. When evaluated by implanting DACB into nude mice, there were no signs of necrosis or inflammation in the epidermal tissues. The bone repair effect of DACB was verified in vivo using sika deer during the antler growth period as an animal model, and the molecular mechanisms of bone repair were further evaluated by transcriptomic analysis of the regenerated tissues. Our findings suggest that the low immunogenicity of DACB enhances the production of bone extracellular matrix components, leading to effective osseointegration between bone and DACB. This study provides a new reference for solving bone defects.

## 1. Introduction

Bone is a vital organ that plays a crucial role in providing support and locomotion. Trauma, infections, tumors, and degenerative diseases can result in bone defects. While small defects, such as fractures, can self-heal, larger defects beyond a critical size require bone substitutes to facilitate effective healing [1]. Various types of bone graft materials are currently employed, including autografts, allografts, xenografts, ceramic-based bone graft substitutes, and polymer-based bone graft substitutes [2]. Each type has its own advantages and disadvantages. Autografts remain the gold standard for bone repair, but are constrained by the limited availability of bone at the donor site, necessitating additional surgeries, increasing surgical morbidity, and raising healthcare costs [3]. Allografts or xenografts come from a wide range of sources and are histologically similar to human bone, but with immune rejection [4]. With careful design, bioceramic materials and synthetic scaffold materials can be 3D-printed to mimic the tiny pores of human bone and provide space for cell adhesion and growth. Nevertheless, the use of these materials for bone tissue repair presents several challenges, including poor mechanical conductivity, suboptimal physicomechanical properties, high brittleness, limited resorption, absence of osteoinductive signals, and uncontrolled and unpredictable in vivo degradability [5,6,7].

The unique physiological microstructure of natural bone tissue is impossible to replicate with bioceramic and polymer replacement materials [8]. The development of bone tissue materials of xenogeneic origin overcomes the obstacles of autologous bone grafting and fulfills the requirements of an ideal bone graft material [9]. Therefore, elimination of immunogenic substances from xenograft materials is essential for the development of xenograft substitutes. Infectious and immunogenic pathogens in bone can be completely removed by high-temperature calcination, but the collagen components in the calcined bone matrix will be seriously destroyed. Additionally, the calcined natural bone material is transformed into inorganic hydroxyapatite and tricalcium phosphate, similar to the extensively studied β-tricalcium phosphate (TCP). This results in significantly reduced bioabsorbability compared to natural bone [10,11,12]. The decellularization technique provides a solution for xenograft bone graft materials [13,14]. The use of decellularized bovine cancellous bone matrix and porcine bone matrix for bone defects has been extensively studied. The results showed that the decellularized cancellous bone material fused completely with the original tissue without causing an inflammatory reaction and exhibited good bioactivity and biocompatibility [15,16,17]. However, the use of decellularized antler cancellous bone material (DACB) has rarely been reported.

Antlers represent the only complex organ in mammals capable of complete cyclic regeneration, occurring in pairs as bone tissue on the skulls of males in most cervine species [18]. The fascinating process begins in spring when old ossified antlers are cast, and new antlers initiate growth from permanent bony prominences called pedicles. During summer, the highly vascularized velvet tissue covering the antler, along with abundant blood supply from the bases, leads to rapid antler elongation (up to 2 cm per day) [19]. By late fall, antlers become fully calcified, and during winter, calcified antlers remain securely attached until the following spring, triggering a new round of regeneration [20,21]. Ossified antlers are used as display organs during the rutting season and as weapons in male combat, and their mechanical properties are reasonably stiff and remarkably tough—well suited for the requirements of bone repair materials [22]. In addition, antlers are similar to human bones in terms of chemical composition and physiological structure [23,24]. The cancellous bone marrow portion of the ossified antler retains only the cancellous bone structure and lacks blood and adipose tissue, facilitating the decellularization process.

The cyclic regeneration of antlers provides a relatively reasonable time span, and the ample blood supply in antlers is a crucial factor for facilitating rapid bone tissue repair and regeneration. He et al. inserted dental implants using sika deer (*Cervus nippon*) antlers as an implant bed and found that the bone remodeling around the implant exhibited biomechanical properties similar to those of human bone. Animal sacrifice was avoided during the experiment [25]. Traditionally, the animal models used to study bone defect repair are mainly rats, rabbits, and goats [26,27,28]. However, these animals face several challenges during experiments, such as fewer blood vessels in the bone and complications of surgical manipulation, and the need to sacrifice animals to obtain experimental samples [29]. Therefore, the use of sika deer antler models for bone repair studies may be an attractive option.

In this study, we employed a combination of physical and chemical methods to prepare DACB. We then fully characterized DACB and investigated its effects on promoting osteogenic differentiation of C3H10T1/2 cells in vitro, along with assessing immunogenicity and angiogenic capacity in vivo. Subsequently, deer antlers during the rapid growth period were selected as an animal model to investigate histocompatibility and bone repair mechanisms. The primary objective of this study was to determine the effectiveness of the improved decellularization method in fabricating DACB and to evaluate whether DACB could exhibit comparable or superior bone repair-promoting properties compared to ceramic TCP scaffolds.

## 2. Materials and Methods

All experiments were performed in accordance with the guidelines and study protocols of the Animal Ethics Committee of Jilin Agricultural University (AEC 2021101103). XKC was acquired from Beijing Xinkangchen Medical Technology Development Co., Ltd. (Beijing, China). TCP was acquired from Shanghai Bio-lu Biomaterials Co., Ltd. (Shanghai, China).

### 2.1. Preparation of DACB

The antlers were harvested in mid-November when they were completely ossified, and were provided by the Jilin Province Zhenyuan deer industry Co., Ltd. (Changchun, China). Cancellous bone from the antlers was obtained by removing the cortical bone from the ossified surface. The acquired cancellous bone material was segmented into strips measuring 20 mm in length, 5 mm in width, and 5 mm in thickness. These strips could also be further divided into other bone pieces tailored to the size of the specific bone defect. The material was treated physically and chemically to remove immunogenic material from the tissue of the bone strips. The physical treatment involved freezing the segmented bone strips at −80 °C for 12 h, followed by thawing in a water bath at 37 °C. This freeze–thaw process was repeated five times to rupture cell membranes through repeated cycles. The material underwent additional chemical treatment, including soaking in 5% hydrogen peroxide (Aladdin, Shanghai, China) for 24 h, followed by rinsing with sterilized water. Subsequently, the samples were treated with a 4% SDS-containing ionic detergent for 24 h, washed repeatedly with sterilized water, and then placed in a 1% Triton X-100 (Aladdin, China) solution. The bone strips were treated for 12 h on a shaker at a constant speed (180 rpm), washed with sterilized water, and then subjected to a 12 h treatment on a shaker at 37 °C using 50 U/mL of DNase I (Macklin, Shanghai, China) and 1 U/mL of RNase (Macklin, China). DNase I facilitated the degradation of nucleic acid sequences, contributing to the removal of immunogenic material. The material was further cleared of antigens using α-galactosidase (α-Gal, Macklin, China) containing 100 U/mL. DACB was soaked in 75% ethanol for 4 h, frozen at −80 °C for 12 h, and then lyophilized for 24 h (ALPHA 1-2 LD plus, Christ, Osterode, Germany). The samples were then sterilized with 25 kGy cobalt-60 (Gammacell^®^ 220, MDS-Nordion, Ottawa, ON, Canada) radiation after sealed packaging for future use. For histological analysis, some samples were preserved in 4% paraformaldehyde (Aladdin, China), with native antler cancellous bone matrix (NACB) samples serving as controls.

### 2.2. DNA Residue Detection and Characterization of Collagen

To assess the degree of decellularization, residual DNA in natural or decellularized tissues was examined using 4′,6-diamidino-2-phenylindole (DAPI, Beyotime, Shanghai, China) staining and hematoxylin and eosin (H&E) staining. DAPI staining involved fixing the scaffold material in 4% paraformaldehyde for 30 min and applying DAPI staining solution for 5 min, followed by observation using a fluorescent microscope. The material was further fixed in 4% paraformaldehyde for 24 h, decalcified using a 10% formic acid (Sigma, St. Louis, MO, USA) solution, transparently dehydrated through a gradient of alcohol, paraffin-embedded, and then sectioned into 5 μm-thick tissue sections. These sections were subjected to H&E staining. For collagen fibronectin staining: collagen fibronectin in DACB was stained using a Masson trichrome staining kit (Solarbio, Beijing, China) and the staining was observed under the microscope. Total DNA was isolated with a Animal Tissues/Cells Genomic DNA Extraction Kit (Solarbio, Beijing, China) according to the manufacturer’s instructions. The concentration of DNA in the extracts was measured by a BioTek Synergy HTX (BioTek, Winooski, VT, USA) plate reader.

### 2.3. Surface Observation and Chemical Composition

The surface microstructure of the scaffold material was examined using a scanning electron microscope (SEM, SU-70 UHR Schottky Analytical FE-SEM, Tokyo, Japan). DACB, NACB, XKC, and TCP were subjected to gold plating on the scaffold surface in a vacuum coater before microstructure observation with the scanning electron microscope. Energy-dispersive X-ray spectroscopy (EDS) was employed for surface element analysis, and chemical analysis of the samples was conducted at a magnified × 60 field of view. EDS mapping was performed in three different regions of each sample. Fourier-transform infrared (FTIR) spectroscopy (Nicolet 6700 FT-IR spectroscopy, ThermoFisher Scientific, Waltham, MA, USA) was utilized to analyze the presence of chemical bonds and structural differences in the different materials (DACB, NACB, XKC, TCP). Pore size and distribution on the material surface were determined using Nano Measurer 1.2.5. XRD measurement was performed on a Rigaku D/MAX-2250 V at Cu Kα (λ = 0.154056 nm, Tokyo, Japan) with a scanning rate of 10° min^−1^ in the 2θ range of 10–80°.

### 2.4. Compressive Strength

A universal mechanical testing machine (CMT5305, Shenzhen New Sansi Material Testing Company, Shenzhen, China) was employed to assess the compressive strength of the scaffold material. The scaffold material was shaped into small pieces measuring 5 mm × 5 mm × 5 mm. These pieces were positioned on the platform of the mechanical testing machine, and measurements were taken at a compression rate of 1 mm/min, applying continuous pressure to the scaffold until fracture occurred. Each group of 5 scaffold materials underwent repeated testing.

### 2.5. Preparation of Extracts for In Vitro Cell Assay

The scaffold material was immersed in DMEM (Gibco, Grand Island, NY, USA) at a concentration of 0.1 g/2 mL and allowed to stand for 24 h at 37 °C in a thermostat to obtain extracts from the scaffold material. The resulting extracts were filtered using a 0.22 μm-filter membrane and stored at 4 °C. For cell culture, 10% fetal bovine serum (FBS, Corning, NY, USA) and 100 U/mL penicillin (Sigma, St. Louis, MO, USA) were added.

### 2.6. Cell Culture

C3H10T1/2 cells were obtained from Shanghai Zhongqiao Xinzhou Biotechnology Co., Ltd. (Shanghai, China). The cells were cultured in DMEM containing 10% FBS and 100 U/mL penicillin in a cell culture incubator at 37 °C with 5% CO_2_. The medium was changed every 3 days. Cells were passaged at a ratio of 1:3 using 0.25% trypsin-EDTA when cell confluence reached 90%. The cells used in the experiments were all below the ninth generation.

### 2.7. Cell Activity Assay on Scaffold Material

To visualize the effect of scaffold material on cell activity, the scaffolds were grouped and placed in 24-well plates. A 50 μL cell suspension was added to the surface and incubated for 30 min. The cell inoculation density was 1 × 10^4^ cells per scaffold. Subsequently, 1 mL of medium was added to each well to completely immerse the scaffold. The cells were stained with a calcein–PI cell viability/cytotoxicity assay kit (Beyotime, Shanghai, China) after 7 days. To investigate the impact of scaffold materials on cell proliferation, cells were seeded on a 96-well plate at a density of 5 × 10^3^ cells/well and cultured for 1, 3, 5, and 7 days in basal culture media or different extract media containing 10% FBS. Cytotoxicity was assessed using the Cell Counting Kit 8 (CCK-8, Biosharp Biotechnology, Shanghai, China). After a 1.5 h reaction in a 5% CO_2_ incubator, absorbance was measured by a microplate analyzer at 450 nm wavelength. Additionally, cells were stained with the calcein–PI cell viability/cytotoxicity assay kit (Beyotime, Shanghai, China) after 7 days. Live cells exhibited green fluorescence, while dead cells exhibited red fluorescence under a fluorescence microscope. Cell counting was performed using ImageJ2 (U.S. National Institutes of Health, Bethesda, MD, USA).

### 2.8. Alkaline Phosphatase (ALP) Staining

Cells were seeded in 24-well plates at a density of 2 × 10^4^ per well. After 24 h of culture, the medium was aspirated and discarded. The experimental group was replaced with osteogenic-induced differentiation medium containing different extract media, while the control group was replaced with normal osteogenic-induced differentiation medium (100 nM dexamethasone, 50 mg/mL L-ascorbic acid 2-phosphate, and 10 mM β-glycerophosphate). The medium was changed every 2 days. After 7 and 14 days of culture, ALP staining and quantitative analysis were performed.

ALP staining followed the instructions of the BCIP/NBT alkaline phosphatase display kit (Beyotime, Shanghai, China). Cells were washed three times with PBS after 7 and 14 days of culture, fixed in 4% paraformaldehyde for 15 min, and then washed again with PBS. Subsequently, 500 μL of BCIP/NBT staining solution was added to the wells, and the cells were stained for 24 h at room temperature under light-proof conditions before observation and photography under the microscope. Staining images were obtained using a microscope (Leica, DMi8, Wetzlar, Germany). Quantification of the ALP-positive area images was performed using ImageJ2 software (National Institutes of Health, Bethesda, MD, USA).

### 2.9. Calcium Deposition Capacity Test

Cells were seeded in 24-well plates at a density of 2 × 10^4^ cells/well. After 24 h of culture, the medium was aspirated and discarded. The experimental group was replaced with osteogenic-induced differentiation medium containing different extract media, while the control group was replaced with normal osteogenic-induced differentiation medium. The medium was changed every 2 days. After 14 and 21 days of culture, the cells were assayed for their calcium deposition capacity using alizarin red staining (ARS).

### 2.10. Osteogenic Differentiation-Related Gene Expression Assay

To investigate the impact of scaffold material on osteogenesis-related gene expression levels in C3H10T1/2 cells, cells were seeded into 24-well plates at a density of 2 × 10^4^ cells/well. After 7 and 14 days of osteogenic induction, total RNA was extracted from the cells using TRIzon Reagent (Cwbio, Beijing, China). Reverse transcription was performed with the PrimeScript™ RT reagent kit with gDNA Eraser (Takara, Dalian, China) to generate cDNA templates. The resulting cDNA was used to assess the expression of key genes involved in constitutive bone differentiation, including osteocalcin (OCN), alkaline phosphatase (ALP), collagen type I (Col I), runt-related transcription factor 2 (Runx2), and osteopontin (OPN). Primer sequences are listed in Table 1. Glyceraldehyde-3-phosphate dehydrogenase (GAPDH) primers were used for normalization. The reaction mixture, containing TB Green Premix Ex Taq™ (Takara, Dalian, China), underwent PCR on a qTOWER384G (Analykit Jena AG, Thuringia, Germany). Data analysis was performed using the 2^−ΔΔCt^ method.

### 2.11. Whole-Body Acute Toxicity Test

Following the standards outlined in GB/T16886 for the systemic acute toxicity testing of medical biological materials [30], NACB and DACB were placed in saline at a concentration of 0.1 g/mL. The solutions were left to stand for 72 h in a constant-temperature chamber at 37 °C to obtain NACB and DACB extracts. Thirty Balb/C mice were randomly divided into three groups, NACB, DACB, and control, with an equal distribution of males and females in each group. The mice were intraperitoneally injected at a dose of 50 mL/kg, with the experimental group receiving the material extract and the control group receiving normal saline. After injection, the mice were kept in separate cages. Toxic reactions were evaluated cording to the indicators of the degree of animal reaction in Table 2.

### 2.12. In Vitro Hemolysis Test

The hemolysis rate of a test material, as per ISO 10993 [31], determines its compliance with safety standards for medical biomaterials. If the hemolysis rate is less than 5%, the material meets the standard; if it exceeds 5%, the material is considered hemolytic and does not meet safety testing standards. In the experimental procedure, fresh rat blood was centrifuged, obtaining red blood cells that were washed and diluted to a 2% concentration. The experiment involved DACB, normal saline and 1% Triton groups, each with three tubes. A 1.5 mL centrifuge tube was used, adding 1 mL of material extract and 0.5 mL of diluted erythrocyte suspension. The tube was incubated for 1 h at 37 °C, 130 r/min on a shaker and photographed. The tubes were then centrifuged at 2000 r/min for 5 min, 100 μL of supernatant was taken in a 96-well plate, the absorbance value (545 nm) was measured using a multifunctional enzyme marker, and the hemolysis rate was calculated according to the following formula:Hemolysis rate HR%=OD(samples)−OD(normal saline)OD(Triton group)−OD(normal saline)×100%

### 2.13. Subcutaneous Implantation

To assess the histocompatibility of DACB in vivo, it was implanted subcutaneously in a nude mouse model, with NACB used as a control group. Cubes of scaffold material measuring 5 mm × 5 mm × 5 mm were prepared and implanted subcutaneously in nude mice to observe their physical health. Tissue samples were collected 21 days after implantation for histological and immunohistochemistry (IHC) examinations. IHC was performed on the subcutaneously grafted scaffold material using α-smooth muscle actin (α-SMA, Abcam, Shanghai, China) as a primary antibody (1:1000) and an UltraSensitiveTM SP (Mouse/Rabbit) IHC Kit (MX Biotechnologies, Fuzhou, China) to evaluate the generation of blood vessels in the DACB after subcutaneous grafting.

### 2.14. Surgical Procedures

Four 3-year-old male sika deer were selected and anesthetized during the rapid growth period of the antlers (after the antlers had developed main beams and brow tines). The surface of the antlers was sterilized with iodophor and alcohol, and then a 5 cm-long incision was made at the root of the antler with a scalpel in the longitudinal direction, and the antler skin was separated with a curved hemostat to expose the bony part of the antler. A 10 mm diameter and 10 mm depth defect were made in the antler body using a hand drill with a 10 mm diameter drill bit. DACB was prepared and implanted into the defect, and the incision was sutured. A control group and TCP group were set up at the same horizontal height, with an interval of 2 cm between each group. Antlers were sampled and analyzed 1 month and 2 months after surgery. Four samples were set up for two sika deer at each time point.

### 2.15. Micro-CT Analysis

All the specimens were scanned using a Skyscan 1172 micro-CT scanner (Bruker, Kontich, Belgium). After standardized reconstruction using NRecon 1.6.10.1 software, the datasets were analyzed using CTAn software (Bruker micro-CT, Antwerp, Belgium). Data of new bone volume ratio (bone volume/total volume, BV/TV), trabecular thickness (Tb.Th), and trabecular spacing (Tb.Sp) were obtained by CTAn1.15.4.0 software.

### 2.16. Histological Evaluation

The extracted bone samples underwent the following procedures. They were fixed in 4% paraformaldehyde for 3 days and then decalcified with 10% formic acid. Subsequently, the specimens were dehydrated sequentially with ethanol and embedded in paraffin. They were then sliced into 5 μm sections using a microtome (Leica, Wetzlar, Germany), followed by staining with H&E as well as Masson’s trichrome method to assess collagen fibers. Finally, they were photographed under a microscope (Precipoint M8, Munich, Germany).

### 2.17. Transcriptome Sequencing

One month after surgery, total RNA was isolated from regenerated tissues of both the DACB and control groups using the Trizol reagent (Invitrogen Life Technologies, Carlsbad, CA, USA), following the manufacturer’s instructions. Subsequently, the concentration, quality, and integrity of the extracted RNA were determined using a NanoDrop spectrophotometer (Thermo Scientific, Waltham, USA). The extracted RNA was then sent to Shanghai Personal Biotechnology Co., Ltd. (Shanghai, China) for library preparation and RNA sequencing (RNA-seq).

### 2.18. Statistical Analysis

The results are presented as means ± SD. Statistical significance was evaluated using GraphPad Prism 9.0.0 (GraphPad Software, La Jolla, CA, USA) software. Statistical analysis for the comparisons of multiple variables was performed using two-way ANOVA followed by Tukey’s post hoc test for multiple comparisons. Student’s *t*-test was used to compare pairs of variables. All experiments were performed in triplicate. Statistical significance was set at *p* < 0.05.

## 3. Results

### 3.1. Preparation and Characterization of DACB

#### 3.1.1. Morphologies of DACB

The outside of the ossified antler consisted of a hard layer of cortical bone, and there was a large amount of cancellous bone at the center. Visually, the ossified antlers retained a three-dimensional porous structure, with some blood stains remaining in the central area of the NACB, giving the central area a brown color (Figure 1A). The cancellous bone was obtained by removing the cortical bone layer, and after physical and chemical decellularization, the cancellous bone scaffold had a white appearance, with large amounts of porous structure visible to the naked eye arranged in a regular longitudinal pattern, with interconnected voids (Figure 1B). DAPI staining revealed that there were many intact nuclei that were readily observable in the NACB (Figure 1C). However, most cells had been removed from the DACB (Figure 1D). Furthermore, H&E staining showed that the NACB possessed a large number of cells, both in and around the extracellular matrix (Figure 1E), but few nuclei were retained in the DACB (Figure 1F). The decellularization of the bone blocks and retained ECM of the bone tissue were evaluated by Masson trichrome staining. The cells and collagen contained in the native antler cancellous bone before decellularization were stained (Figure 1G), whereas the DACB showed no cells; however, all the collagen was well preserved (Figure 1H), indicating that the decellularization process did not significantly alter the ECM or collagen of the bone tissue. Quantification results (dsDNA) in initial dry weight NACB and DACB showed that the double-stranded DNA (dsDNA) content in the DACB (6.36 ± 2.4 ng/mg) was significantly lower, whereas the dsDNA content (135.47 ± 9.6 ng/mg) in the NACB was much higher (*p* < 0.001) (Appendix A).

#### 3.1.2. SEM Analysis

The porous microstructures of NACB and DACB were examined phenotypically by SEM, and the pore size of the scaffold materials was measured using a scale. SEM results showed the porous surface structure present on the materials (Appendix A). The overall microstructure of DACB is very similar to that of NACB (Figure 2A), indicating that the decellularization process did not severely alter the ultrastructure of cancellous bone (Figure 2B). Finer cancellous bone trabeculae were seen on the surface of XKC, a commercially available scaffold material (Figure 2C), and the pores of TCP were generated by very round wax beads with a range of specific sizes (Figure 2D). The average pore diameters of NACB, DACB, XKC, and TCP scaffold were about 369.43 μm, 337.81 μm, 779.15 μm, and 392.54 μm, respectively (Figure 2E–H).

#### 3.1.3. Energy-Dispersive X-ray Spectroscopy (EDS) Findings

The results of surface elemental analysis showed that the cancellous bone materials mainly contained the following elements: C, O, P, and Ca (Appendix A). Additionally, DACB and NACB contained the element Mg (Figure 2I–K), whereas TCP was primarily composed of Ca and P (Figure 2L). As shown in Table 3, analysis of the weight percentages of Ca and P elements in NACB, DACB, XKC, and TCP indicated that the Ca/P ratios of the various materials were 2.02, 1.78, 1.96, and 1.90, respectively.

#### 3.1.4. FTIR and XRD Analysis

To investigate the differences in the chemical compositions and material structures before and after decellularization, FTIR analysis was performed: 556 cm^−1^ and 600 cm^−1^ were the bending vibration absorption peaks of PO_4_^3−^ in hydroxyapatite, 871 cm^−1^ was the bending vibration absorption peak of CO_3_^2−^, 1009 cm^−1^ was the symmetric stretching absorption peak of PO_4_^3−^, 1411 cm^−1^ and 1457 cm^−1^ were the stretching vibration absorption peaks of CO_3_^2−^, and 1534 cm^−1^ was the amide. The DACB and NACB peaks were in the same position, indicating that no new material appeared during the decellularization process. Since XKC and TCP did not contain an organic collagen fiber chemical structure, mainly phosphate and carbonate, they did not have a 1534 cm^−1^ amide II band N-H bending vibration characteristic absorption peak. Therefore, these results further demonstrated that the prepared DACB had both inorganic and organic chemical composition and structures (Figure 3A). The XRD results showed that TCP had high crystallinity and clear crystal structure. In contrast, the crystallinity of the natural bone matrix (XKC and DACB) was much lower. It is noteworthy that the decellularization process had little effect on the crystallinity of the NACB samples, which mainly maintained an amorphous or low-crystallinity structure (Figure 3B).

#### 3.1.5. Mechanical Properties

The mechanical properties of the different materials were evaluated by maximum compressive strength. Values for the XKC group (4.42 ± 0.71 MPa) and TCP group (7.67 ± 0.83 MPa) were significantly lower than the other groups (*p* < 0.05). There was no significant difference in mechanical properties between the NACB (13.56 ± 1.93 MPa) and DACB groups (12.11 ± 0.89 MPa) (*p* > 0.05), which proved that there was no significant effect on the scaffold biomechanical properties through the decellularization process (Figure 3C).

### 3.2. In Vitro Cell Studies

#### 3.2.1. Assays of Cell Viability and Proliferation

Cell viability studies showed that DACB, XKC, and TCP materials support cell growth and proliferation. Further, direct and indirect methods were used to evaluate the biocompatibility of the materials. In the direct method (materials co-cultured with cells), most of the cells showed green fluorescence (live cells) and a few showed red fluorescence (dead cells) after 7 days of incubation based on live/dead cell staining results (Figure 4A). In the indirect assay, cell proliferation in the presence of the material extract (conditioned medium) was comparable to that of the complete medium (control), indicating that cell growth and proliferation were not inhibited when exposed to any of the material extracts (Figure 4B–D). These data suggest that the porous nature of the materials provides environmental conditions for cell adhesion, colonization, and proliferation and permits an adequate supply of nutrients.

#### 3.2.2. Osteogenic Differentiation and Related Gene Expression of C3H10T1/2

ALP is a marker of early to mid-stage differentiation of osteoblasts. The level of ALP in each group gradually increased with the increase in culture time: the DACB group stained darker than the XKC, TCP, and control groups at 7 days of culture (Figure 5A). The trend of staining status at 14 days of incubation was the same as that at 7 days. The results of quantitative analysis of staining showed that there was no significant difference in ALP activity in the control, XKC, or TCP groups (*p* > 0.05), while the ALP in the DACB group was significantly higher than that of other groups (*p* < 0.05) (Figure 5B).

Calcium deposition is a marker of osteoblast differentiation and maturation. The chromogenic level of calcium nodule deposition gradually increased with increasing culture time in each group. At 14 days of culture, the number of alizarin red-stained cells was significantly higher in the DACB group compared with the control, XKC, and TCP pairs. The trend of staining status at 21 days of culture was the same as that at 14 days. Among the cells with calcium nodule formation after 21 days of culture, the number of stained calcium nodules in the DACB group was significantly higher than that of the other groups (Figure 5C). Thus, the induction effect of DACB on the osteogenic differentiation ability of the cells was further demonstrated.

To further elucidate the impact of scaffolds on C3H10T1/2 differentiation, we utilized RT-qPCR to examine the expression of relevant genes. The results demonstrated a significant upregulation of osteogenesis-related genes, including ALP, OCN, OPN Col I, and Runx2, in the DACB group on day 14 compared to the XKC and TCP groups (*p* < 0.001) (Figure 5D–H). These findings strongly indicate that DACB enhances in vitro osteoinduction.

### 3.3. Biosafety of DACB

#### 3.3.1. Systemic Acute Toxicity Test

The systemic acute toxicity test was performed by observing whether the animals showed signs of toxicity after intraperitoneal injection of the implanted materials so as to judge the magnitude of toxicity. General observations were that mice in the NACB group showed severe toxicity and significant weight loss 1, 2, and 3 days after injection, while mice in the DACB and normal saline groups showed good vital signs and no obvious toxicity (Figure 6A). The results of histology showed that the liver, kidney, spleen, heart, and lung tissues of DACB group were intact and similar to those of saline group, and no necrosis, inflammation, apoptosis and degeneration were observed, while in the NACB group, the red marrow and white boundary of spleen tissue were blurred, the splenic sinus was atrophied, and the number of red blood cells located in the red marrow was significantly reduced (Figure 6B). Therefore, the systemic acute toxicity test indicated that the DACB prepared in this study is biologically safe.

#### 3.3.2. In Vitro Hemolysis Test

In vitro hemolysis testing was performed to evaluate the effect of the implanted materials on blood by detecting the hemolysis rate of red blood cells after rupture. As shown in Figure 6, erythrocytes in both the DACB and normal saline groups were deposited at the bottom of the centrifuge tube without significant hemolysis, while there was significant hemolysis in the 1% Triton group (Appendix A). According to the calculation formula, the hemolysis rate of the extract of the DACB group was less than 5% of the requirement (Figure 6C). The experimental results demonstrate that the DACB prepared in this study has no significant harm on blood and meets the safety standards for medical biomaterials.

#### 3.3.3. Biocompatibility In Vivo

To further evaluate the histocompatibility of DACB, in vivo histocompatibility experiments were performed via subcutaneously transplanting DACB into nude mice for 21 days. The NACB group exhibited poorer healing of the surgical site, inflammatory exudate, and a significant decrease in animal activity (Figure 7A). In contrast, there were no significant changes in the implantation site of the DACB group, and there were no adverse reactions such as exudate, pus, or fistulae at the surgical site. Additionally, there were no significant changes in activity of the animals before and after implantation (Figure 7C). The results of gross tissue examination showed that in the NACB group, there was a large number of dilated capillaries around the implant, and the surface was encapsulated by a layer of yellowish inflammatory adherent connective tissue (Figure 7B), whereas in the DACB group, there were no significant inflammatory adhesions or connective tissue at the tissue surrounding the implant (Figure 7D). Histologic examination showed that in the NACB group, there was a large number of necrotic and fragmented inflammatory cells around the implant (Figure 7E). In the DACB group, the tissue retained its original structure, the pores were filled with fibrous tissue, and there was no inflammatory cell infiltration (Figure 7F). IHC analysis of α-SAM demonstrated the presence of neovascular tissue in DACB, revealing vascular regeneration 21 days after implantation (Figure 7G). Overall, the results of subcutaneous implantation of DACB show very good biosafety and biocompatibility.

#### 3.3.4. Micro-CT Reconstruction and Quantification

In order to further verify the bone repair ability of DACB (Figure 8A), TCP was chosen as the positive control group (Figure 8B) to repair a bone defect with a diameter of 10 mm on a regenerating sika deer antler (Figure 8C). At 1 month and 2 months after surgery, the central region of the bone defect in the control group still showed obvious bone defects accompanied by inflammatory exudate, the scaffold material in the TCP group had fused with the surrounding tissues, but the undegraded TCP scaffold material was still clearly visible in the defect. In the DACB group, the stent material fused well with the surrounding tissue, and the repaired tissue was similar to the surrounding tissue (Figure 8D). Micro-CT examination showed that the DACB group showed abundant new bone formation throughout the bone defect area, and the osseointegration was relatively complete and continuous at both edges of the defect (Appendix A). The scaffold material in the TCP group was poorly absorbed and retained its original structure 2 months after implantation. Regenerated bone tissue was also observed in the control group, but significant defects remained in the central region of the bone defects (Figure 8E). The quantitative analysis of BV/TV, Tb.Th, and Tb.Sp further supported the optimal bone regeneration in the DACB group (Figure 8F).

#### 3.3.5. Histological Analysis

Histologic evaluation of the repaired tissues was performed using H&E staining and Masson staining. According to the results of H&E staining, the control group showed newly formed bone tissue at the edge of the defect area, but the middle part of the defect area was filled with fibrotic tissue exhibiting inflammation and a fibrotic healing pattern. In the TCP group, the boundary between the bone defect and the cement was clearly visible, and the edge of the defect did not have trabecular-like structures. There was almost no new bone formation within the cement, but the number of cells infiltrating the scaffold gradually increased with increasing implantation time. However, compared with the other groups, the DACB group showed significant scaffold degradation and new bone formation, the regenerated bone tissue fused better with the surrounding tissues, and new trabeculae filled the middle and edges of the bone defects. The thickness of regenerated trabeculae was similar to that of the surrounding normal bone tissue 2 months after operation, and the bone matrix was mature (Figure 9A). Further use of Masson staining for collagen formation was used to assess the maturity of the regenerated bone tissue, with the newly generated collagen tissue appearing blue, while the mature bone and other material appeared red. The regenerated bone tissue in the DACB group 2 months after operation showed the formation of mature plate-like bone (red staining). Collagen expression, the number of bone trabeculae, and the number of mature bone marrow cavities were higher in the DACB group compared with the control and TCP groups (Figure 9B).

#### 3.3.6. RNA-Sequencing Analysis

To further explore the mechanism by which DACB promotes bone tissue repair, the repaired tissues were subjected to RNA-seq analysis. PCA was used to assess the consistency of the three biological replicates as well as the differences in transcriptome profiles between the two repaired tissue types. The results demonstrated that the reproducibility of RNA-seq data was high among the three biological replicates in the control and DACB groups, indicating significant principal component differences in the samples between the groups (Figure 10A). DESeq analysis identified a total of 324 differentially expressed genes (DEGs) between the control and DACB groups (*p* < 0.05), of which 124 were up-regulated (red) and 200 were down-regulated (blue) DEGs (Figure 10B). Clustering analysis of DEGs showed significant changes in the transcriptome profiles of the control and DACB groups (Figure 10C). To analyze the function of DEGs, GO and KEGG pathway enrichment analyses were performed. Analysis of GO biological process, cellular components and molecular functions of RNA-seq data showed that up-regulated genes were mainly enriched in extracellular matrix organization, extracellular matrix, and collagen beaded filament (Appendix A). Down-regulated genes were mainly enriched in the B cell receptor signaling pathway, inflammatory response, calcium ion-regulated exocytosis, immune response activation, and other molecular functions, as well as immune response-activating signal transduction and regulation of inflammatory response at the cell surface (Appendix A). For the up-regulated genes in the KEGG analysis results, the most enriched KEGG pathways were ECM–receptor interaction, focal adhesion, PI3K–Akt signaling pathway, axon guidance, Wnt signaling pathway, and Hippo signaling pathway (Appendix A). For the down-regulated genes, the most enriched KEGG pathways were the B cell receptor signaling pathway, glycerolipid metabolism, cGMP–PKG signaling pathway, fat digestion and absorption, and hematopoietic cell lineage (Appendix A).

## 4. Discussion

Bone grafting is crucial in mending segmental bone defects. In this study, we fabricated DACB through a decellularization process. Upon analyzing the composition and structure of DACB, the results showed that DACB retained the 3D structure and collagen components of cancellous bone tissue. The in vitro experiments demonstrated that DACB exhibited a pronounced ability to promote osteogenic differentiation in C3H10T1/2 cells. In the bone repair animal model, DACB showed a significant osteogenic differentiation-promoting ability compared to the TCP scaffold material, highlighting DACB’s significant potential for promoting new bone regeneration. Therefore, our study provides a new reference for the clinical treatment of bone defects.

Decellularization removes cells and immunogenic substances from bone tissue while preserving the natural ECM components [32,33]. However, some damage to the ECM can occur during the decellularization process. To better preserve the more intact ECM scaffolds, it is essential to maintain homeostasis between maintaining the ECM structure and removing cellular components [34]. Therefore, effective decellularization should maximize the removal of cellular components and genetic material while minimizing ECM damage, thus preserving the bioactivity of the ECM, 3D ultrastructure, and specific biomechanical properties of the ECM [35]. In this study, we were able to reduce the use of chemicals and the preservation of ECM ultrastructure by the ice crystals produced in the cells during freeze–thaw cycling to penetrate the cell membrane and disrupt the cells [36]. Further, the cytogenetic material and cellular debris released during the freeze–thaw process can be washed with SDS to fully comply with the standard requirements of removing cells and eliminating at least 90% of DNA [37]. Finally, our quantitative analysis of dsDNA in DACB based on the principle of confirming effective decellularization reported by Crapo et al. showed a content of 6.36 ± 2.4 ng/mg on a dry weight basis, which complies with the regulatory requirement of less than 50 ng/mg [38,39]. Masson staining showed that the collagenous component of the ECM was preserved during the decellularization process. As in previous studies [40,41,42,43], subcutaneous implantation of DACB into nude mice showed good histocompatibility and neovascularization. Further allografting of DACB for the repair of bone defects showed that no immune reaction or inflammation occurred during the repair process.

Regarding the results showing that repair using DACB was superior to TCP, we propose that this is related to the graft material. In contrast to TCP tissue-engineered scaffolds, decellularized bone tissue matrix can provide a natural bone matrix component for bone repair [44]. This matrix mainly retains the ECM on the surface of the bone matrix after decellularization. In addition, other inorganic elements such as magnesium, which can promote bone repair, are also retained [45]. Therefore, DACB offers the following advantages for bone tissue regeneration. (1) The ECM in DACB promotes rapid integration of the clot formed at the defect, creating a favorable microenvironment for endogenous cell recruitment and tissue regeneration. This facilitates cell adhesion and differentiation [37,46]. (2) Due to the large amount of growth factors deposited in the ECM during rapid antler regeneration, the biological behavior of the cells in the microenvironment is coordinated, which promotes osteoblast differentiation and induces proliferation [47]. (3) DACB retains the three-dimensional structure of cancellous bone, which provides a space for the orderly growth of new bone tissue and improves the quality of regenerated bone tissue. (4) The interconnected, multidimensional pore structure of DACB provides numerous binding sites for cell membrane receptors, which are key to determining and maintaining cellular functions [48,49]. This is the main reason that the repair effect of the DACB group was significantly better than that of the other two groups in the process of bone defect repair.

To understand the biological mechanism of DACB in the repair of antler bone defects, we selected regenerated bone tissue from the control and DACB groups one month after surgery for RNA-Seq analysis. The functional enrichment analysis results of the differentially expressed genes demonstrated that the implantation of DACB up-regulated gene expression, effectively activating osteogenesis-related signaling pathways and biological processes. This activation promoted cellular osteogenic differentiation, ECM secretion, and inward vascular growth, positively regulating osteogenic signaling pathways and biological processes [50]. Among the down-regulated genes, the B cell receptor signaling pathway is a series of signals produced by B cells in response to antigenic stimulation. It stimulates B cell differentiation to generate an immune response that helps the body to fight against pathogens [51]. The downregulation of the B cell receptor signaling pathway further suggests that DACB implantation reduces biological processes associated with inflammatory responses in the organism. These results are consistent with histological findings.

In selecting an experimental animal model, we chose antlers during the growth period to study bone defect repair. The study found that there was no need to sacrifice animals to obtain experimental samples. Additionally, the antlers on the stag’s forehead simplified the experimental process. The rich vascular environment during antler growth provided nutrients to the defect site and facilitated the rapid transformation of the study material [19]. Furthermore, the ossification process of antlers is relatively fast, usually completing within 3–4 months after the initiation of growth [52]. Therefore, antler tissue during the growth period may be an ideal animal model for studying bone defects, especially for investigating the transformation and calcium deposition capacity of bone repair materials.

While our preliminary studies have shown the low immunogenicity and bone repair ability of DACB in allografts, additional validation in other xenogeneic animal models is crucial to strengthen the findings. Our next steps involve extending our research to various xenogeneic models to obtain more comprehensive insights. Additionally, we plan to leverage tissue engineering principles for bone defect repair. Future investigations will focus on exploring the synergistic effects of DACB in combination with cells and growth factors, expanding our research across a broader spectrum of animal models.

## 5. Conclusions

In summary, this study successfully prepared biologically active DACB scaffolds through a decellularization process. DACB demonstrated osteogenic microenvironment and mechanical properties comparable to natural bone tissue, supporting bone formation. In vitro studies revealed that DACB promoted cell adhesion, proliferation, and differentiation. Moreover, DACB exhibited low immunogenicity in vivo, indicating excellent biocompatibility compared to NACB. Implantation of DACB into 10 mm-sized bone defects in growing antlers accelerated new bone formation compared to TCP and control. Transcriptome analysis highlighted the upregulation of genes associated with coordinated cellular activities in bone regeneration, particularly involving the classical ECM–receptor interaction and Wnt signaling pathway. Overall, the results suggest that DACB prepared by removing immunogenic material holds promise as a bioactive material for supporting bone remodeling and other tissue regeneration applications.

## Figures and Tables

**Figure 1 biomolecules-14-00907-f001:**
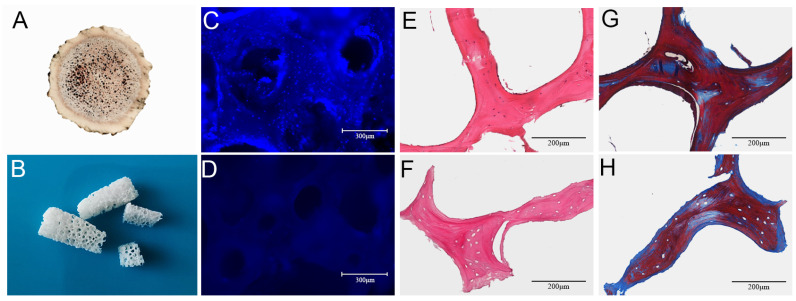
Morphology and histology of NACB and DACB. Gross appearance of NACB (**A**) and DACB (**B**). DAPI staining shows NACB (**C**) has a large number of cell nuclei. After decellularization, few cell nuclei were observed on DACB (**D**). H&E staining shows a lot of cells around and inside NACB (**E**), but few cells on DACB (**F**) were seen. Masson’s trichrome staining shows collagen fibers in NACB (**G**) and DACB (**H**).

**Figure 2 biomolecules-14-00907-f002:**
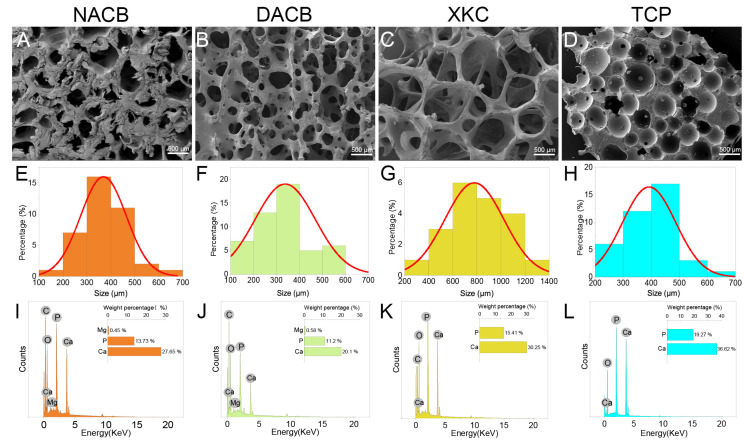
Material characterization. (**A**–**D**) Surface morphology of different scaffolds was observed using SEM. (**E**–**H**) Histogram of the pore size distribution determined by Nano Measurer 1.2.5 software. (**E**) The pore size of NACB was mainly 300–400 μm. (**F**) The pore size of DACB was mainly 300–400 μm. (**G**) The pore size of XKC was mainly 600–800 μm. (**H**) The pore size of TCP was mainly 400–500 μm. (**I**–**L**) Elemental analysis of different scaffolds was conducted using EDS.

**Figure 3 biomolecules-14-00907-f003:**
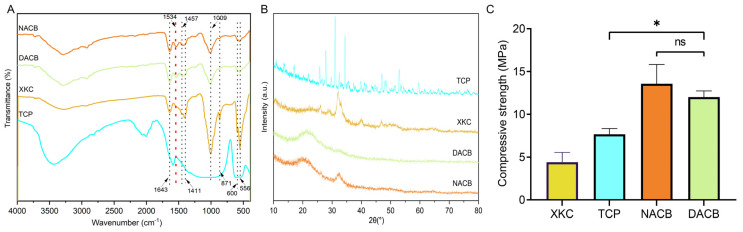
FTIR and compressive strength of materials. (**A**) FTIR spectra of NACB, DACB, XKC and TCP. Red dotted line: absorption peaks of amide II. (**B**) XRD spectra of different material. (**C**) Compressive strength of different materials. (ns, *p* > 0.05; *, *p* < 0.05).

**Figure 4 biomolecules-14-00907-f004:**
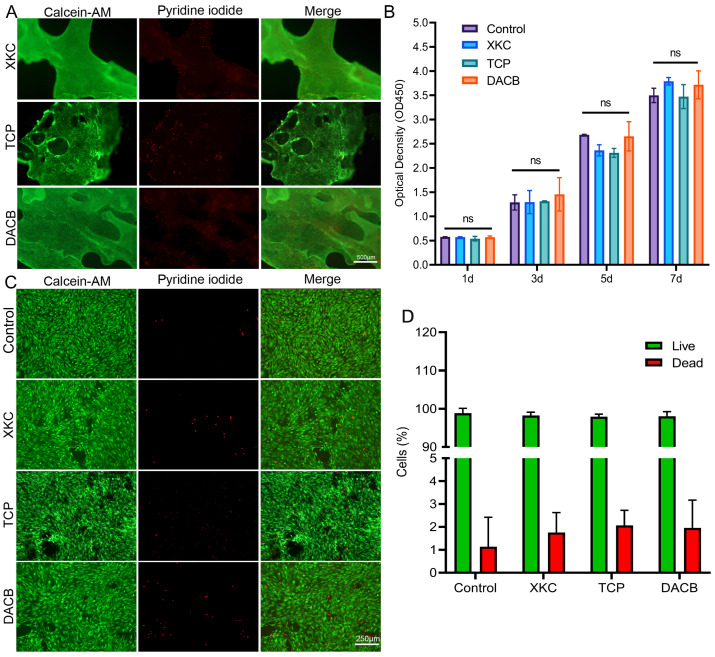
Cell activity and proliferation of C3H10T1/2 cultured on materials. (**A**) Calcein AM (green, live cells) and PI (red, dead cells) staining for C3H10T1/2 on XKC, TCP and DACB at day 7. (**B**) The cell proliferation of C3H10T1/2 cells cultured on extracts from different materials was assessed using the CCK-8 method. (**C**) C3H10T1/2 cells cultured on different material extracts were measured using the calcein–PI cell viability method at day 7. (**D**) Live/dead cell count of C3H10T1/2 cells cultured on different material extracts on the 7th day. (ns, *p* > 0.05).

**Figure 5 biomolecules-14-00907-f005:**
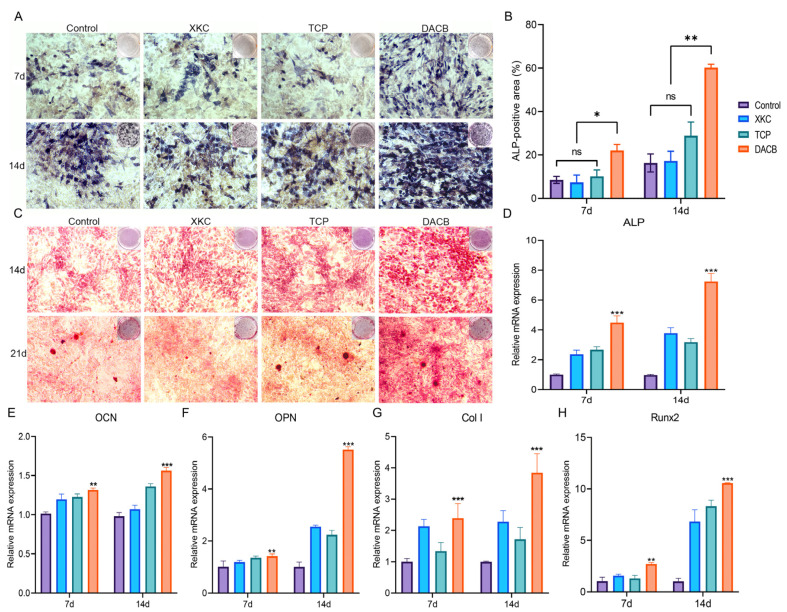
DACB promotes osteogenesis in vitro. (**A**,**B**) ALP staining and quantitation assay after 7 and 14 days of incubation. (**C**) Alizarin red staining on days 14 and 21 after osteogenic induction. (**D**–**H**) RT-qPCR quantification for the mRNA expression of ALP, OCN, OPN, Col I and Runx 2 in cells cultured for 7 and 14 days. (ns, *p* > 0.05; *, *p* < 0.05; **, *p* < 0.01, and ***, *p* < 0.001).

**Figure 6 biomolecules-14-00907-f006:**
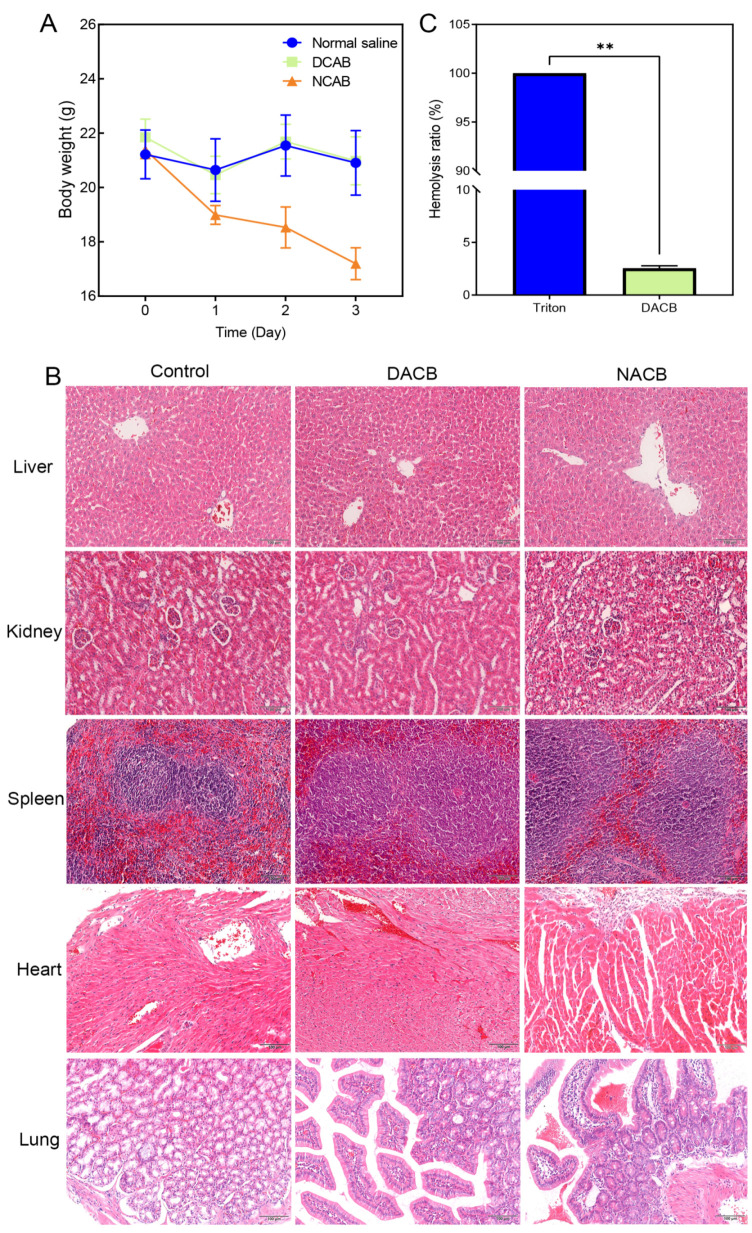
Biosafety of DACB. (**A**) Body weight. (**B**) Biosafety was evaluated by H&E staining of the liver, kidney, spleen, heart, and lung. (**C**) Hemolysis rate. (ns, *p* > 0.05, **, *p* < 0.01).

**Figure 7 biomolecules-14-00907-f007:**
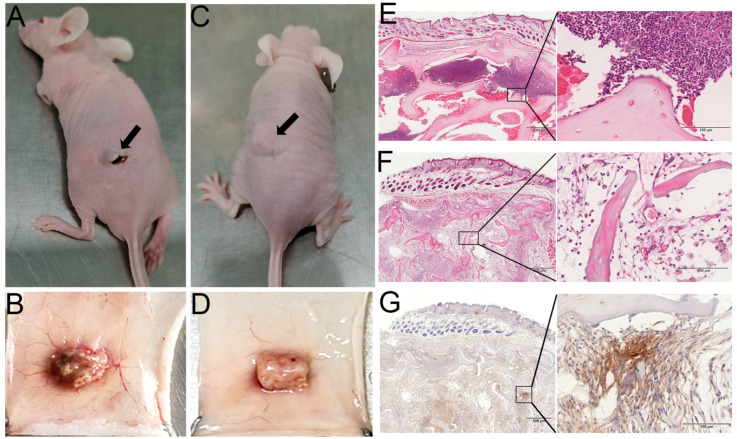
Histocompatibility and angiogenesis of DACB in nude mice. Gross morphological observations of NACB (**A**,**B**) and DACB (**C**,**D**) at 21 days after transplantation into nude mice subcutaneously (arrows). H&E staining results of NACB (**E**) and DACB (**F**) at 21 days after transplantation into nude mice subcutaneously. (**G**) IHC staining of α-SMA. Scale bar: low magnification = 500 μm; high magnification = 100 μm.

**Figure 8 biomolecules-14-00907-f008:**
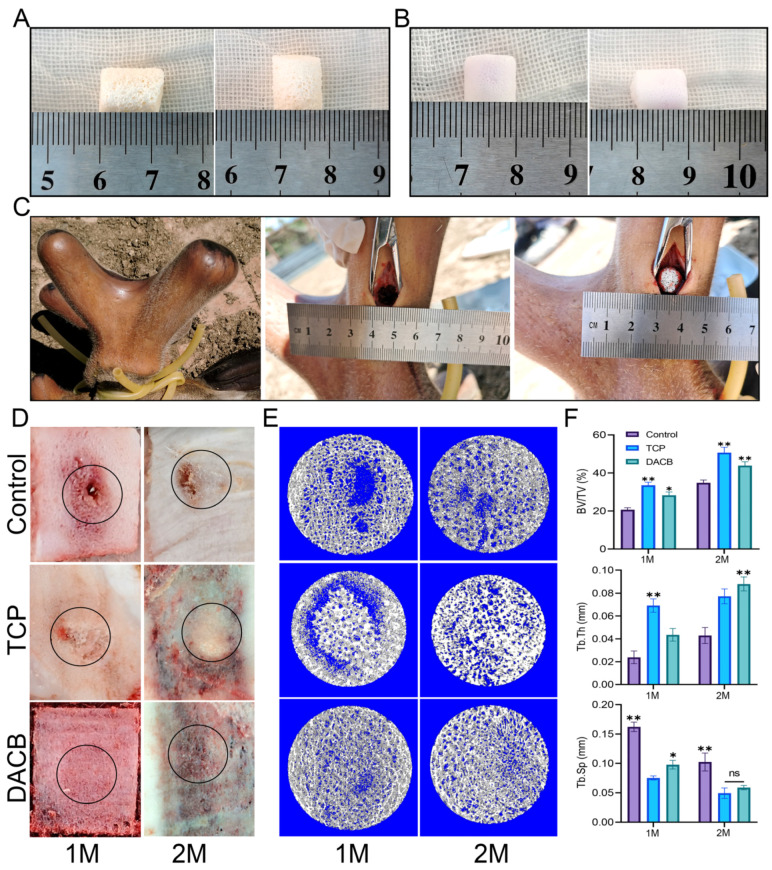
DACB promotes bone regeneration of antler bone defects. (**A**) DACB material. (**B**) TCP material. (**C**) Construction of bone defect model and exhibition of material implant surgery process. (**D**) Macroscopic evaluation of repair tissue at 1 and 2 months. Black circles indicate the defect area. (**E**) Three-dimensional reconstructed micro-CT images at 1 and 2 months after surgery. (**F**) Quantitative analysis of BV/TV, Tb.Th, and Tb.Sp based on µCT examination. (ns, *p* > 0.05; *, *p* < 0.05, and **, *p* < 0.01.)

**Figure 9 biomolecules-14-00907-f009:**
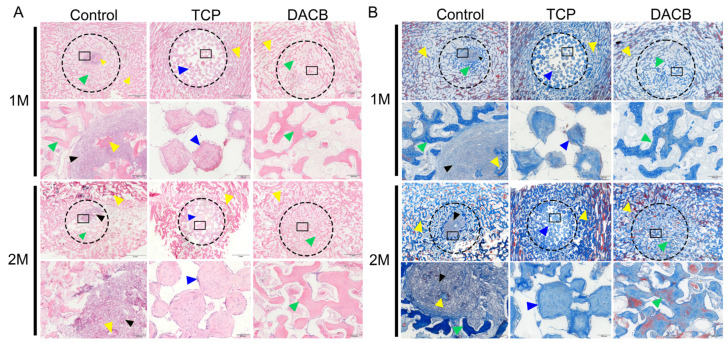
Histological evaluation of repair tissue by H&E (**A**) and Masson staining (**B**) at 1 and 2 months after surgery. The host bone is marked with a yellow triangle, the graft is marked with a bule triangle, the fibrous tissue is marked with a black triangle, and the new bone is marked with a green triangle. Dotted circle is the bone defect area, and the black box is at high magnification. Scale bar: low magnification = 2 mm; high magnification = 200 μm.

**Figure 10 biomolecules-14-00907-f010:**
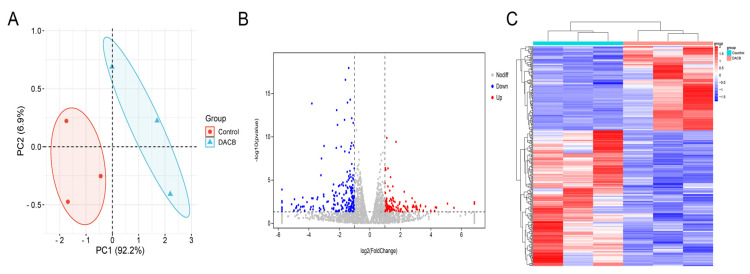
The mRNA-seq and differentially expressed gene (DEG) analyses comparing the repaired tissue between the control and DACB groups at 1 month after surgery. (**A**) PCA of mRNA-seq. (**B**) Volcano map of DEG in control and DACB groups. (**C**) Clustering map of DEG between control and DACB groups.

**Table 1 biomolecules-14-00907-t001:** Primer sequences used for RT-qPCR analysis.

Osteogenic Gene	Primer Sequence (5–3′)
ALP	Forward CCAGAAAGACACCTTGACTGTGG Reverse TCTTGTCCGTGTCGCTCACCAT
OCN	Forward GCAATAAGGTAGTGAACAGACTCC Reverse CCATAGATGCGTTTGTAGGCGG
OPN	Forward GCTTGGCTTATGGACTGAGGTC Reverse CCTTAGACTCACCGCTCTTCATG
Col I	Forward TGGAGAGAGCATGACCGATG Reverse GAGCCCTCGCTTCCGTACT
Runx2	Forward CCTGAACTCTGCACCAAGTCCT Reverse TCATCTGGCTCAGATAGGAGGG
GAPDH	Forward CATGGCCTTCCGTGTTCCTA Reverse GTTGAAGTCGCAGGAGACAAC

**Table 2 biomolecules-14-00907-t002:** Criteria for evaluation of systemic acute toxicity test.

Reaction Level	Evaluation Criteria
Non-toxic	Animals were in good general condition and showed no obvious signs of intoxication.
Mild toxicity	Experimental animals moved normally, but exhibited mild respiratory distress and signs of abdominal irritation.
Obvious poisoning	Experimental animals showed difficulty in breathing, severe abdominal irritation, reduced activity and food intake, and mild weight loss.
Severe intoxication	Experimental animals exhibited cyanosis, tremors, respiratory failure, etc., and significant weight loss.
Death	Experimental animals died after injection.

**Table 3 biomolecules-14-00907-t003:** Energy-dispersive X-ray spectroscopy (EDS).

Elements	Chemical Compositions (wt%)			
	NACB	DACB	XKC	TCP
P	13.73 ± 3.57	11.20 ± 2.42	15.41 ± 1.22	19.27 ± 2.18
Ca	27.65 ± 6.69	20.10 ± 5.28	30.25 ± 8.45	36.62 ± 4.34
Mg	0.45 ± 0.13	0.58 ± 0.21	——	——
Ca/P	2.02	1.78	1.96	1.90

## Data Availability

All data generated or analyzed during this study are included in this published article. RNA-seq datasets have been uploaded to the Sequence Read Archive (SRA) repository under the BioProject code PRJNA1090160 (https://www.ncbi.nlm.nih.gov/) accessed on 24 March 2024.

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
