# Peer review of "Decellularized Antler Cancellous Bone Matrix Material Can Serve as Potential Bone Tissue Scaffold"

_biomolecules, 2024, doi:10.3390/biom14080907_

Round 1
Reviewer 1 Report
Comments and Suggestions for Authors
The manuscript "Decellularized Antler Cancellous Bone Matrix Material Can Serve as Potential Bone Tissue Scaffolds" provides a thorough investigation into the utility of decellularized antler cancellous bone matrix (DACB). The experiments were rigorously conducted, with a clear presentation of interesting and valuable results. The limitations of this study were also fairly written. The reviewer believes that the manuscript can be considered for publication in Biomolecules after addressing the following few revisions.
Comment #1
The authors presented in vivo effects of DACB in comparison with "control" and TCP (Figures 8, 9, 10). The definition of this control group is not clearly stated. The reviewer suggests that the authors clearly define the control group.
Comment #2
The authors compared DACB with a TCP scaffold but did not mention the crystalline phase of DACB. The reviewer recommends that the authors discuss the crystalline phase of DACB by citing previous articles or providing XRD data.
Comment #3
The results of the acute toxicity tests are clearly presented. However, according to the Materials and Methods section 4.11, the toxic reactions should be evaluated based on the criteria (1 - 5). The reviewer recommends that the authors provide a table summarizing the evaluation on the toxic reactions.
Comment #4
The hemolysis results (Figure 6C) are confusing. According to the equation defined in the manuscript, the hemolysis rate of normal saline should be defined as 0%. However, the result shows a hemolysis rate greater than 0% for normal saline. The reviewer recommends that the authors clarify this definition.
Comment #5
Minor comments.
1) Some of abbreviations (e.g., NACB, PCA) needs to be defined at first use.
2) The unit "Mpa" in the main text should be corrected to "MPa".
3) Statistical analyses for multiple comparisons cannot be performed using ANOVA, and usually performed using post-hoc test such as Tukey's method.
4) In the Introduction, Page 2, Line 58: "In contrast to the current market practice of eliminating zoonotic infectious and immunogenic pathogens present in bone through high-temperature calcination." This sentence needs grammatical correction.
5) Sample specification:
Materials and Methods section
4.5, "The scaffold material was immersed in DMEM at a concentration of 0.1 g/2 mL"
4.7, "The scaffolds were grouped and placed in 24-well plates"
The dimensions of scaffolds need to be specified.
6) Equation for hemolysis rate: "ODValue(samlpes)" should be corrected to "OD (sample)".
7) Results 2.3.2, Page 8, Line243: "significant hemolysis in the 1% Triton group (fig S2)" is likely to be corrected to "... (Figure S3)".
Author Response
Comment 1: The authors presented in vivo effects of DACB in comparison with "control" and TCP (Figures 8, 9, 10). The definition of this control group is not clearly stated. The reviewer suggests that the authors clearly define the control group.
Response: Thank you for pointing this out. Our experimental groups were as follows:
Control group: Not filled with any scaffold material.
Positive control group: Filled with TCP material.
Comment 2: The authors compared DACB with a TCP scaffold but did not mention the crystalline phase of DACB. The reviewer recommends that the authors discuss the crystalline phase of DACB by citing previous articles or providing XRD data.
Response: Thank you for pointing this out. We agree with this comment. We have added XRD data to the article. Line: 159-161, 378-382.
Comment 3:The results of the acute toxicity tests are clearly presented. However, according to the Materials and Methods section 4.11, the toxic reactions should be evaluated based on the criteria (1 - 5). The reviewer recommends that the authors provide a table summarizing the evaluation on the toxic reactions.
Response: Thank you for pointing this out. We have made tables. See Table 2 in the article.
Comment 4: The hemolysis results (Figure 6C) are confusing. According to the equation defined in the manuscript, the hemolysis rate of normal saline should be defined as 0%. However, the result shows a hemolysis rate greater than 0% for normal saline. The reviewer recommends that the authors clarify this definition.
Response: We agree with this comment. We have made adjustments to Figure 6C. Line:464.
Comment 5: Minor comments.
1) Some of abbreviations (e.g., NACB, PCA) needs to be defined at first use.
Response: Thank you for pointing this out. We agree with this comment. We've made changes in the manuscript article.
2)The unit "Mpa" in the main text should be corrected to "MPa".
Response: Thank you for pointing this out. We agree with this comment. We've made changes in the manuscript article. e.g.: line 385, 387.388.
3) Statistical analyses for multiple comparisons cannot be performed using ANOVA, and usually performed using post-hoc test such as Tukey's method.
Response: Thank you for pointing this out. We agree with this comment. We have made changes in "2.18. Statistical analysis".
4) In the Introduction, Page 2, Line 58: "In contrast to the current market practice of eliminating zoonotic infectious and immunogenic pathogens present in bone through high-temperature calcination." This sentence needs grammatical correction.
Response: Thank you for pointing this out. We agree with this comment. We've made changes in the Introduction.
5) Sample specification:
Materials and Methods section
4.5, "The scaffold material was immersed in DMEM at a concentration of 0.1 g/2 mL"
4.7, "The scaffolds were grouped and placed in 24-well plates"
The dimensions of scaffolds need to be specified.
Response: Thank you for pointing this out. We were not able to measure the exact size of the dimensions during the experiments because during the experiments we found that the TCP and XKC materials are highly brittle and cannot be prepared into fixed shaped scaffold materials for the experiments, so during the experiments we used irregular granular materials for the above experiments.
6) Equation for hemolysis rate: "ODValue(samlpes)" should be corrected to "OD (sample)".
Response: Thank you for pointing this out. We agree with this comment. We've made changes in the manuscript article. Line: 260.
7) Results 2.3.2, Page 8, Line243: "significant hemolysis in the 1% Triton group (fig S2)" is likely to be corrected to "... (Figure S3)".
Response: Thank you for pointing this out. We agree with this comment. We've made changes in the manuscript article.
Reviewer 2 Report
Comments and Suggestions for Authors
In the following manuscript Yusu Wang, et al. are regarding about decellularized antler cancellous bone matrix material. The material is proposed as promising for bone tissue scaffold.
It is research article composed of Abstract, Introduction, Results, Discussion, Materials and methods, and Conclusions.
The introduction is extensive and closely related to the topic. The authors cited 50 references and most of them is from last 20 years, what makes the manuscript current and closely related to current trends in tissue engineering.
Generally the manuscript is well organized, the experiment is very interesting, and the research is very advanced, But I have some comments about the article.
1. The discussion have to be shortened. Additionally, some parts of discussion should be the part of Introduction. I mean e.g. lines 335-377. It is closely related to the topic, but most of the cited examples about known experiment should be described in Introduction.
2. Details about chemicals should be presented (e.g. company and country of company). e.g: line 531, 522, 520, 564,
3. Why C3H10T1/2 cells have been chosen to this experiment?
4. Line 524: Were the materials frozen first? For how long? What exactly device was used for freeze-drying? (more details about equipment, and parameters should be presented)
5. Line 525: What exact device was used for sterilization? (more details about equipment, and parameters should be presented)
Thank you
Author Response
Comment 1: The discussion have to be shortened. Additionally, some parts of discussion should be the part of Introduction. I mean e.g. lines 335-377. It is closely related to the topic, but most of the cited examples about known experiment should be described in Introduction.
Response:Thank you for pointing this out. We agree with this comment. We have revised the discussion.
Comment 2: Details about chemicals should be presented (e.g. company and country of company). e.g: line 531, 522, 520, 564,
Response:Thank you for pointing this out. We agree with this comment. We've made changes in the manuscript article. e.g: line 118, 121, 124, 127….
Comment 3: Why C3H10T1/2 cells have been chosen to this experiment?
Response:Because this cell is a precursor cell to osteoblasts, their final differentiated form is the osteoblast. The purpose of our experiment was to test whether our substance had a bone-contributing effect in isolation, so this cell was chosen.
Comment 4: Line 524: Were the materials frozen first? For how long? What exactly device was used for freezedrying? (more details about equipment, and parameters should be presented)
Response:Thank you for pointing this out. We agree with this comment. We have made modifications in the article. “Line 127: …frozen at -80°C for 12 h and then lyophilized for 24 h (ALPHA 1-2 LD plus, Christ, Ger-many).”
Comment 5: Line 525: What exact device was used for sterilization? (more details about equipment, and parameters should be presented)
Response:Thank you for pointing this out. We agree with this comment. We have made modifications in the article. “Line 129: … with 25kGy cobalt-60 (Gammacell® 220, MDS-Nordion, Canadian)….”
Reviewer 3 Report
Comments and Suggestions for Authors
1. The authors present the development of a Decellularized Antler Cancellous Bone Matrix Material that can serve as potential bone tissue scaffolds, thereby enhancing the relevance of this research in the field of bone scaffolds. I believe the study is suitable for publication with some suggested improvements.
2. #2Could the authors provide higher quality microstructure images through SEM investigation, including higher magnification scale for all images (BIO as well)?
3. In this investigation, DACB was engineered for potential applications in osteogenesis and bone repair. The authors present a decellularized antler bone matrix. I believe the chemical surface should be analyzed by XPS before and after decellularization. Can the authors provide more information on this analysis?
Author Response
Comment 1: The discussion have to be shortened. Additionally, some parts of discussion should be the part of Introduction. I mean e.g. lines 335-377. It is closely related to the topic, but most of the cited examples about known experiment should be described in Introduction.
Response:Thank you for pointing this out. We agree with this comment. We have revised the discussion.
Comment 2: #2Could the authors provide higher quality microstructure images through SEM investigation, including higher magnification scale for all images (BIO as well)?
Response: Thank you for pointing this out. We have provided higher magnification SEM images in the Supplementary file (Fig. S2). For the other results, we thought that the magnification used in the article could explain our results, so higher magnification pictures of the other results were not prepared.
Comment 3: In this investigation, DACB was engineered for potential applications in osteogenesis and bone repair. The authors present a decellularized antler bone matrix. I believe the chemical surface should be analyzed by XPS before and after decellularization. Can the authors provide more information on this analysis?
Response:Thank you for your valuable feedback and suggestion regarding our manuscript. We believe that the EDS analyses in the article adequately address the questions about the changes in surface chemistry.
Round 2
Reviewer 3 Report
Comments and Suggestions for Authors
The article can be published in this form. In the future, I recommend considering that EDX is not a reliable analysis for investigating surface chemistry. Ineligible magnification scales of images, etc., are unprofessional.